# Toxicometabolomics: Small Molecules to Answer Big Toxicological Questions

**DOI:** 10.3390/metabo11100692

**Published:** 2021-10-09

**Authors:** Ana Margarida Araújo, Félix Carvalho, Paula Guedes de Pinho, Márcia Carvalho

**Affiliations:** 1Associate Laboratory i4HB, Institute for Health and Bioeconomy, Faculty of Pharmacy, University of Porto, 4050-313 Porto, Portugal; felixdc@ff.up.pt (F.C.); pguedes@ff.up.pt (P.G.d.P.); 2UCIBIO—Applied Molecular Biosciences Unit, REQUIMTE, Laboratory of Toxicology, Department of Biological Sciences, Faculty of Pharmacy, University of Porto, Rua de Jorge Viterbo Ferreira nº228, 4050-313 Porto, Portugal; 3FP-I3ID, FP-ENAS, University Fernando Pessoa, Praça 9 de Abril, 349, 4249-004 Porto, Portugal; 4Faculty of Health Sciences, University Fernando Pessoa, Rua Carlos da Maia, 296, 4200-150 Porto, Portugal

**Keywords:** toxicometabolomics, drug toxicity, toxicity pathways, biomarkers, prediction models

## Abstract

Given the high biological impact of classical and emerging toxicants, a sensitive and comprehensive assessment of the hazards and risks of these substances to organisms is urgently needed. In this sense, toxicometabolomics emerged as a new and growing field in life sciences, which use metabolomics to provide new sets of susceptibility, exposure, and/or effects biomarkers; and to characterize in detail the metabolic responses and altered biological pathways that various stressful stimuli cause in many organisms. The present review focuses on the analytical platforms and the typical workflow employed in toxicometabolomic studies, and gives an overview of recent exploratory research that applied metabolomics in various areas of toxicology.

## 1. Introduction

Within the advancement of science and technology in recent years, the application of omics strategies has allowed a paradigm shift that provides a holistic perspective on biological studies. Among the available omics sciences, metabolomics has stood out [1], and is increasingly being used in the study of several toxic agents—a subfield named toxicometabolomics—in order to better understand their toxicity mechanisms, as well as to identify new biomarkers and target organs. This review focuses on recent advances in toxicometabolomics research and discusses some of the challenges and pitfalls encountered in metabolomics work.

## 2. What Is the Metabolome?

The term ‘metabolome’ was first used by Steven Oliver in the late 1990s [2], and was defined as the complete set of low-molecular-weight compounds (<1500 Da) present within biological systems, and their interactions. Its composition is affected by the upstream influence of the genome, transcriptome, and proteome, as well as by environmental and lifestyle factors, drugs, and/or underlying diseases [3,4]. Although the full number of metabolites present in humans is not yet known, due to the high complexity of the metabolome, the Human Metabolome Database (HMDB; http://www.hmdb.ca, version 4.0, 2018) contains over 114,000 metabolite entries (peptides, lipids, amino acids, nuclei acids, carbohydrates, and organic acids, among others) with a wide dynamic concentration range, from high abundance (>1 mM) to relatively low abundance (<1 nM) [5].

Some authors restrict the metabolome to the set of endogenous metabolites [6,7,8], although the exogenous metabolites (e.g., drugs and body microbiota) also play an important role in an organism’s physiology or pathophysiology [9]. In fact, the metabolome can be divided into four categories: (i) the intracellular metabolome (or endometabolome), which includes all metabolites produced by each cell type, tissue, or organism [10,11]; (ii) the extracellular metabolome (or exometabolome), which refers to the metabolites secreted or consumed by the cells [11,12]; (iii) the microbial metabolome produced by the microbiota [10,13]; and (iv) the xenometabolome, which includes the metabolites derived from xenobiotics, pollutants, and diet [10,14].

## 3. Metabolomics: Concept and Strategies

The study of metabolic profiles of a given organism and the changes in that same profile was called ‘metabonomics’ by Nicholson et al. [15] and ‘metabolomics’ by other authors [8]. These two terms are often used interchangeably, although each one has a specific purpose. The two fields employ similar analytical and data processing, and have a common goal, metabolome analysis; however, while metabolomics intends to identify and quantify all metabolites (endogenous and exogenous) present in a specific biological sample [8], metabonomics studies how the metabolic profile of a complex system changes in response to specific stimuli, such as a disease or treatment [15]. In this review, for simplicity, the term ‘metabolomics’ will be used regardless of the purpose under study.

According to the central dogma of molecular biology (Figure 1), deoxyribonucleic acid (DNA) is transcribed into ribonucleic acid (RNA), which is then translated into proteins, the activities of which result in the formation of myriad metabolites [3]. Metabolites are the ultimate product of all regulatory complexity present in the cell, tissue, or organism [3], and therefore are the most proximal molecules of the biochemical activity that occur in an organism in response to physiological and pathophysiological stimuli [16]. Thus, considering the metabolome as most predictive of phenotype [17], metabolomics is the omics science that provides the most functional information [18].

Besides offering a powerful holistic approach to understand biological processes, metabolomics also has several other benefits over lower-level approaches (genomics, transcriptomics, and proteomics) (Table 1). First, it is able to provide information about the specific area of metabolism that is affected [3]. As the downstream product of gene expression, the metabolome can provide amplified alterations when compared to transcriptome or proteome, increasing the sensitivity to detect biochemical changes [19]. This approach is of higher throughput, generally lower in cost, and provides data that is much less complex and more informative than any other omics technology [3]. It is also important to note that metabolomics facilitates the translatability of the results from experimental models to humans, since, regardless of the organism’s complexity, the chemical structures of the metabolites are universal [19]. Despite all these advantages, a key limitation to metabolomics is the fact that metabolites do not have a direct link to the genome, as many genes may determine the synthesis and turnover of a single metabolite, which makes it difficult to interpret metabolomics data [11,20]. Another drawback compared to transcriptome and proteome is the wide variety of physicochemical properties of the metabolites, which make them more differentially extractable and determined by one single analytical platform [20]. Additionally, the identification and quantification of many metabolites at the same time often makes it difficult to include the data in a physiological context that matches to the current understanding of metabolism. Still, this can be valuable in the discovery of hitherto unknown pathways [11].

Within metabolomic studies, three analytical strategies can be distinguished: the global or untargeted approach, the metabolic profiling, and the targeted approach (Figure 2) [8,22]. Untargeted metabolomics is based on studying the largest number of metabolites as analytically possible, without having a priori knowledge on the nature and identity of the measured compounds, looking for variations that can be used to discriminate groups of samples [8]. Consequently, these studies are characterized by the production of large and complex amounts of data, which are now surpassed by the use of powerful bioinformatics tools. Given its potential for hypothesis generation and its comprehensive metabolome coverage, untargeted metabolomics is often the first approach taken by researchers looking for a metabolic research question [23]. Metabolic profiling focuses on the quantitative analysis of a set of predefined metabolites that belong to a specific class of compounds (e.g., sugars, amino acids, lipids, or organic acids) or a particular metabolic pathway [8,24]. On the other hand, targeted metabolomic studies are hypothesis-driven experiments and seek to measure a specific set of metabolites [8]. In this type of strategy, a selective sample preparation is usually applied and optimized to quantify the concentrations of metabolites with high precision and accuracy, usually to validate an untargeted analysis [25].

A hypothesis-generating metabolomics approach covers different strategies to provide sample classification (i.e., case/control): (i) fingerprinting; and (ii) footprinting analysis. The metabolic fingerprint refers to the global snapshot of the metabolites present inside the cells (intracellular metabolome or endometabolome), while the metabolic footprinting explores the (exo)metabolites excreted or consumed by an organism (extracellular metabolome or exometabolome), providing a cumulative picture of metabolism over time [11,26,27]. Compared to fingerprinting analysis, footprinting analysis has the advantage of having a relatively simple sampling that does not require extremely time-consuming quenching and extraction steps [25,28]. In addition, it allows the monitoring of the metabolic changes over time in cultures in the same cells [29]. However, the interpretation of extracellular changes may be conditioned by difficulties in establishing direct relationships between the exometabolome and the cellular metabolic state [29].

## 4. Metabolomics Workflow

A good experimental design and the choice of appropriate methods of samples and data processing are prerequisites for the success of any metabolomics study. For the present topic, the workflow commonly used in high-throughput untargeted metabolomic studies (Figure 3) will be described, as well as the main challenges found in each stage.

### 4.1. Biological Question Formulation

The first stage of a metabolomics study consists of the right formulation of the biological question to be answered. This process is of utmost importance, since it will determine the experimental design to be followed, namely the type of approach (untargeted vs. targeted metabolomics), sample type, sample size, experimental conditions to be tested, frequency and timing for sample collection, sampling conditions, storage conditions, sample preparation strategies, and the analytical platforms to be used.

### 4.2. Which Sample to Choose?

The choice of samples for metabolomic studies depends essentially on the research question. Metabolomic experiments are typically carried out in complex matrices, such as cell culture samples (cell extracts and culture media), tissues, and biofluids (urine, whole blood, serum, plasma, feces, seminal fluid, saliva, sweat, breast milk, bile, and cerebrospinal fluid) [28,30]. Cells and tissues are usually used to investigate the mechanisms of action associated with pathophysiological processes, whereas biofluids are studied to identify new biomarkers [28]. For this topic, the advantages and limitations of the type of samples most used in metabolomics research will be discussed.

#### 4.2.1. Cellular Models

In vitro metabolomic studies provide information on specific cell types under different conditions, which may be important for the development of drugs that target specific cell phenotypes [31]. These studies are easy to execute and interpret due to the lack/minimization of confounding factors (e.g., gender, age, and lifestyle factors) [29,31]. On the other hand, in vitro studies can be criticized for being very different from the natural environment, since most cellular systems are reduced to just one type of cell (without cell-cell interaction) kept in artificial conditions [32]. In vitro studies also face some issues of variability derived from growth-medium formulation, number of passages, cell density, quenching, and extraction processes [29,33]. Some of the above-mentioned issues can be solved through an appropriate experimental design and the implementation of standard operating procedures for preanalytical handling of metabolomic samples [34,35].

A broad diversity of cellular models, including tumorigenic and nontumorigenic immortalized cell lines [36,37,38], primary cells obtained from different tissues [39,40], and stem cells [41,42] have been used in in vitro metabolomic studies, access to which is facilitated through cell culture biobanks, such as the American Type Culture Centre (ATCC, www.atcc.org) [36]. Immortalized cell lines offer several advantages over primary and stem cells, as they are economical and highly available, are easy to handle, can be kept in culture for longer periods of time, provide an unlimited and pure population of cells with a stable phenotype that guarantee reproducible results, and circumvent ethical concerns associated with the use of animal and/or human samples [29,31,43]. However, the authenticity of cell lines can become a problem, since a relevant percentage of cell lines, even in biobanks, can be contaminated or erroneously characterized [32].

Primary cells, on the other hand, have the ability to retain the morphological and functional characteristics of their tissue of origin, and constitute the closest model to the in vivo situation [29]. Nevertheless, their low availability (particularly those of human origin), their high phenotypic variability, and the considerable drop in cell viability after isolation limit their widespread use in metabolomic experiments [29].

Stem cells are undifferentiated cells that have long-term capabilities for multipotent differentiation and self-renewal. They have the ability to replenish damaged somatic cells and maintain a self-renewal reservoir of progenitors that is crucial for the homeostasis in many tissues [44]. These cells can be obtained from many sources, by invasive and noninvasive methods, and have the potential to differentiate into several specific cell types. However, they are associated with several limitations in terms of acquisition and isolation, in addition to the fact that the use of some of these cells (embryonic stem cells) is considered unethical under the laws of many countries [45].

More recently, several novel cell culture technologies have become available (for example, co-cultures of different cell types, 3D culture, organ-on-chip, among others). These new models have enormous physiological relevance, as they are also able to generate results closer to the in vivo situation, but they also increase the number of parameters that must be controlled to reduce the variability (e.g., pH, waste and metabolic end-products accumulated in the medium, availability of oxygen and nutrients, and cell size and shape, among others), which can be an obstacle in metabolomics [46].

#### 4.2.2. Tissues

The analysis of tissue samples provides localized snapshots of metabolic activity, making it possible to study the origin of the metabolites, unlike what happens with biofluids that reflect changes in multiple organs [28,47]. In certain situations (for example, in cancer studies), tissues allow a better match between disease and nondisease samples, as it is possible to remove both samples from the same organ, reducing the impact of confounding factors [48]. However, this biological matrix requires special attention, since an important aspect to take into account in tissue analysis is its heterogeneity (for example, due to the presence of different cell types or zones with different oxygenations and enzyme systems), which may introduce additional biological variations [49]. The preparation of tissue samples is very laborious, which can represent a disadvantage for metabolomic studies [50]. Furthermore, tissue samples are usually collected under anesthesia, which can lead to tissue-specific metabolic changes, a problem not found in other types of samples [51].

#### 4.2.3. Urine and Blood

The application of metabolomics to the analysis of biofluids faces different challenges compared to that of cells and tissues. Urine and blood are the two most studied biofluids in metabolomics due to their ease of collection, richness in metabolites, for allowing the study of temporal changes, and for being ethically acceptable and cost-effective [47,52,53]. However, the metabolic profile of these biofluids is modulated not only by diseases or pharmacological/toxicological effects, but also by confounding factors, such as age, gender, body mass index, lifestyle, nutritional status, environmental factors, and gut microbiota, among others, making it difficult to understand causal processes [53,54,55]. In addition, although these samples have the ability to reflect a global metabolic picture, they have a low level of specificity, as they reflect the function of multiple organ systems [28].

Urine, in particular, offers advantages over other biofluids due to its noninvasive nature, being suitable for young children and individuals for whom venous access is problematic [56]. In addition, it provides unlimited volume, simple sample collection and pretreatment, and lower protein content and sample complexity, including fewer intermolecular interactions [57]. On the other hand, compared with whole blood and blood products, urine appears to have a more sensitive metabolic profile, suffering a more evident diurnal variation, greater inter- and intravariability, being also more susceptible to changes due to the possibility of bacterial contamination [55,58]. Blood, plasma, and serum can produce more relevant information, since they are less affected by confounding factors [56], although out of the three, whole blood offers greater metabolic detail and higher reproducibility [59,60]. Relatively small metabolic differences were found between serum and plasma, although serum samples provide greater overall sensitivity due to a higher concentration of metabolites [61]. On the other hand, due to the lack of the clotting step, plasma processing is faster, simpler, and more reproducible [62].

### 4.3. Sample Collection and Preparation

The method of collection and preparation of samples can have a significant impact on the metabolomics data and conclusions derived from a study, since these preanalytical steps can be sources of variation [28]. An inadequate procedure can lead to high variability, interference with instruments, loss of metabolites, or even the formation of degradation metabolites [60]. Thus, although there is no ideal method, these steps must be optimized according to the type of sample chosen and based on a compromise between efficient extraction and minimal loss of metabolites [63].

The first critical factor to consider is selecting an appropriate collection time, because a large fraction of the metabolism oscillates due to circadian rhythm, physical activity, and dietary status [64,65]. Twenty-four-hour sampling (in the case of urine) is preferable to eliminate daytime variability [53], but if sample collection is spread over time, all samples should be collected within the same time period and under similar conditions (e.g., early morning, fasting) [66].

To ensure the stability of biofluids, some additives may be added to the collection tubes, namely sodium fluoride, sodium azide, heparin, citrate, or ethylene diamine tetra acetic acid (EDTA). Considering that these substances can affect the efficiency of extraction and derivatization processes and the ionization process during the mass spectrometry (MS) acquisition, thereby suppressing metabolite ionization and/or introducing artefact signals, its use should be consistent throughout the study and should be adequate to the analytical platform [53,61,67]. For example, for plasma preparation, the choice of anticoagulant addition is critical and should be carefully considered before sample collection, since, although good quality data have been obtained for all additives, the metabolic profile can be strongly influenced. Particularly, EDTA is poorly suited to the analysis of polar metabolites, while the use of citrate compromises the analysis of citric acid and derivatives [61]. Heparin is recommended to be used in ultra-high-performance liquid chromatography-mass spectrometry (UPLC-MS) and nuclear magnetic resonance (NMR) analysis, since it does not cause interference, but it should be avoided in liquid chromatography-mass spectrometry (LC-MS) approaches, in which EDTA is preferable [60].

Regarding tissues, considering their heterogeneity, sampling should always be done in the same organ region [50,60,68]. On the other hand, only specific regions of the organs may respond to certain stimuli, and therefore the analysis of a small sample of tissue obtained from an unaffected region can be misleading. Thus, whenever possible, the whole organ should be analyzed [50,60,68], although this strategy leads to the loss of potentially important spatial information [50]. Advances in MS-based tissue imaging may lead to the development of methods in which the metabolite profiles preserve spatial resolution, but in fact, these methods are still being improved, so they cannot be used in high-throughput mode [50].

An additional procedure to consider during tissue sampling should be the cleaning of samples with saline solutions in order to avoid contamination with blood metabolites [60,69]. The washing step is also critical in cell sampling to remove all the extracellular media components and provide better signal-to-noise ratios [70].

Another crucial point for reducing variability and improving data quality is the quenching step. This procedure aims to interrupt cellular metabolism to prevent the turnover of metabolites such as adenosine triphosphate (ATP) or glucose-6-phosphate, and to obtain a precise picture of the metabolome at the time of sampling. For that, during collection, samples should be kept at the lowest temperature possible, and the metabolism must be stopped immediately after collection in order to avoid possible bias in analysis, and consequently, misleading results [29]. Snap-freezing in liquid nitrogen is the most commonly employed protocol for quenching [50,71,72], although many authors also performed this step using organic solvents adjusted for very cold temperatures and/or extreme pHs [70,73,74,75]. This step is mandatory for metabolomic studies in tissue and cells, but is usually omitted for blood and urine samples due to the fact that the metabolic integrity of these biofluids is maintained after collection for a few hours at 4 °C [61].

Following quenching, the next step is to extract the metabolites. In the case of cell models, according to the study aim, quenching and extraction can be combined or sequential [29,73]. The extraction method must be highly efficient and nonselective, and its choice presents a great challenge due to the heterogeneity of the metabolome. Liquid–liquid extraction is one of the most applied extraction methods [76,77,78], although several other methods can be used (e.g., supercritical fluid extraction [79], solid-phase extraction [80], or solid-phase microextraction [81]). Depending on the planned analysis, a monophasic (such as water/methanol, water/acetonitrile, 100% methanol) or biphasic (water and methanol, often associated with a nonpolar solvent such as chloroform or dichloromethane) solvent solution, in varied proportions and temperatures, can be used for extraction of a large panel of metabolites [29,73,77,82,83]. The use of repeated extraction cycles involving similar or different solvents to those used in the first cycle can improve the extraction efficiency. However, the extraction yield depends not only on the mixture of solvents used, but also on the complete rupture of the cells during extraction, so freeze/thaw cycles, ultrasonication, and homogenization by mechanical means can help increase the effectiveness of the extractive process [29,78]. Even after a highly efficient extraction, some metabolites may be present at a low concentration level. Hence, it is prudent to concentrate samples to reach lower limits of detection during the analytical analysis. Evaporation of the extracted solution to dryness and lyophilization are the most common procedures used to concentrate and preserve the samples [29].

Regarding storage, whenever possible, samples should be aliquoted to avoid repeated freeze–thaw cycles that lead to a progressive change in the metabolic profile and loss of sample quality [50,61]. The thawing steps should always be performed on ice to increase gradually the temperature of the samples [61]. In addition, regardless of the sample under study, a temperature of at least −80 °C is recommended for long-term storage before analysis to prevent metabolite degradation and loss of unstable metabolites [53,60,61].

### 4.4. Analytical Platforms

Advances in analytical techniques in recent years have facilitated metabolomic studies, improving our capacity to obtain more data from biological samples. However, the complexity of the metabolome, due to the diversity of metabolites with differences in molecular weight, polarity, solubility, volatility, and concentrations, challenges the capabilities of any single analytical platform. Thus, the optimal and simultaneous extraction, detection, and quantification of all the metabolites is not possible with a single approach, and to overcome this problem, several analytical platforms can be used to expand metabolite coverage [16,84]. To date, NMR and MS are the most implemented technical approaches to generate metabolomics data.

NMR is a spectroscopic technique that is based on the energy absorption and re-emission of atomic nuclei due to variations in an external magnetic field [16]. In NMR-based metabolomic studies, hydrogen is the most commonly targeted nucleus (^1^H-NMR) due to its natural abundance in biological samples, although other nuclei (e.g., carbon (^13^C), phosphorus (^31^P), or nitrogen (^15^N)) can be used to obtain complementary metabolic information [16]. This technique exhibits a series of favorable characteristics in the study of the metabolome (Table 2, and therefore has been reported in about one-third of recent publications in the area. Briefly, NMR has excellent reproducibility and quantitative accuracy; has the ability to provide structural information; and is fast, nondestructive, cost-efficient, and suitable for high-throughput analysis. However, its limited sensitivity and resolution have restricted the number of metabolites detected, representing a great challenge in the study of complex biological samples [1,16,85,86]. NMR spectroscopy also has the ability to analyze, with no pretreatment, the metabolic profile of intact tissues, cell extracts, and living organisms using high-resolution magic angle spinning (HR-MAS) techniques [86]. In general, one-dimensional (1D) ^1^H-NMR spectroscopy is the most commonly used method for high-throughput metabolomic studies due to its short acquisition time and because it provides a direct measure of metabolite concentration and information on their chemical structure. However, sometimes it is necessary to resort to longer experiments such as two-dimensional NMR (2D-NMR), which are useful to assist in the identification of metabolites and increase their specificity, since this technique allows the separation of overlapping spectral peaks [85].

Due to the evident superiorities of sensitivity (nM to pM range), selectivity, and a wide dynamic range over other techniques, MS has become an ideal platform for metabolomic applications [87]. As recently stated by Wishart et al. [5], MS provides a wider metabolome coverage than NMR, as it has been reported that the average number of metabolites detected by this technique is considerably higher (197 by MS vs. 37 by NMR). Mass spectrometry is used to identify and/or quantify a wide range of analytes using the mass-to-charge (m/z) ratio of ions generated from a sample. All MS techniques require an initial ionization step, and then each molecule generates different peak patterns that define the fingerprint of the original molecule [16].

A wide range of instrumental variants are currently available for MS analysis. Table 2 summarizes the advantages and limitations of some of the most implemented ones in metabolomics. The simplest form of MS is direct infusion-mass spectrometry (DI-MS), which is based on the direct injection of samples into the spectrometer without prior chromatographic or electrophoretic separation. This considerably reduces the analysis time, avoids sample dilution, and improves the repeatability and accuracy, but also results in ion suppression/ion enhancement and low ionization efficiency [63,87]. To reduce ion-suppression problems, MS can be coupled with several separation techniques, such as capillary electrophoresis (CE), liquid chromatography (LC), or gas chromatography (GC) [63]. Each of these techniques has a specific selectivity for certain compounds, providing different information about the composition of samples [63]. 

The separation of compounds in CE is carried out quickly and in a simple way based on their different migrating velocities in the electric field [88]. Although relatively uncommon in metabolomic studies, CE-MS is able to provide useful information about polar and charged metabolites (e.g., amino acids, amines, inorganic ions, nucleotides, and small peptides, among others); however, its low capacity in sample loading and reduced sensitivity limit its application in untargeted metabolomics [88,89].

In contrast, LC and GC are currently the leading separation techniques used in untargeted metabolomics [87]. Simplicity in sample preparation is an advantage of LC, since derivatization is not needed, as occurs with GC [63]. Additionally, LC-MS can handle a large variety of metabolites due to a wide choice of stationary phases that allow a high versatility in metabolome coverage [90]. On the other hand, GC-MS is only suitable for the analysis of volatile compounds or compounds that can be derivatized to volatile forms. In spite of this limitation, its resolution and reproducibility associated with the commercial mass spectral database available can facilitate metabolite identification, one of the main bottlenecks in metabolomics [63,87,88].

High-resolution mass spectrometry (HRMS) is another variant increasingly used to produce metabolomics data. Due to its high mass resolution and mass measurement accuracy, HRMS considerably improves the metabolomic data quality, and is especially beneficial for metabolite identification in complex biological mixtures.

At present, other exciting and novel MS-based metabolomics technologies, such as matrix-assisted laser desorption/ionization (MALDI) mass spectrometry imaging and desorption electrospray ionization (DESI) mass spectrometry imaging, have been developed [87,91]. These techniques use imaging methods to provide in situ spatial information for multiple metabolites simultaneously, while the morphological integrity of the analyzed tissues or cells is maintained [87,91]. Usually, spatial information is still obtained through histological or immunohistochemistry analysis. However, histological staining is non-molecular-specific, and immunohistochemistry requires prior knowledge of the target analytes and is limited to a small number of analytes [87,91]. Although imaging mass spectrometry technologies open a new perspective in the metabolomics field, some analytical challenges still need to be optimized, namely sample preparation and the balance between spatial resolution and sensitivity of analyte detection [91].

### 4.5. Bioinformatics and Statistical Tools in Metabolomics

Metabolomics analysis of biological samples results in an enormous amount of data that can be time-consuming and difficult to process and analyze manually. Therefore, it is necessary to use fast and accurate bioinformatics and statistical tools to deal with the large and complex raw data sets, and provide a workable and understandable format in order to extract meaningful biological information [16,84]. Each of these steps is described in detail below.

#### 4.5.1. Data Preprocessing

After data acquisition, a two-dimensional matrix of intensities is constructed, in which each row corresponds to observations/object (sample, time point, etc.), and each column represents a variable/metabolic feature (e.g., chemical shift, or *m/z*-retention time (RT) pair). It is important to note that instrumental (e.g., chromatographic column degradation) and biological factors (such as pH, ionic strength, and protein content) can affect metabolomics data set by inducing fluctuations in chemical shifts/retention times, intensity, and background noise [94,95]. Therefore, prior to statistical analysis, the matrix needs to be corrected to improve the signal quality and reduce possible bias [16]. Depending on the analytical technique, different preprocessing methods are used, but generally, peak/spectrum alignment, baseline correction, deconvolution, peak detection, normalization, and scaling are considered standard steps [16,84,96,97]. Over time, several commercial and open access software packages, such as MetAlign [98], MZmine [99], and XCMS [100] have been developed to automate these procedures.

Briefly, alignment is one of the main processing steps in metabolomic studies. In both chromatography and NMR, the peaks/spectra can be shifted due to instrumental variations or interferents, which means that throughout the experimental acquisition, the retention time or ppm is not constant for each metabolite [96]. Thus, alignment is required to ensure correct correspondence of peaks/spectra between all samples [101]. 

Baseline elevations due to, for example, bleeding of the chromatographic column also affect metabolomics data, since a true-zero value is shifted upward, so baseline correction should be used to remove those variations [96].

Peak detection allows researchers to identify all the true features present in the chromatogram/spectra and avoid the detection of false positives, and then integrate their areas to provide a (semi)quantification of the underlying metabolite [97]. However, peaks can sometimes overlap in certain areas of the chromatogram/spectrum, which presents a challenge in the analysis and interpretation of data. As a solution, deconvolution techniques can be implemented to extract the intensity of each individual peak [102]. 

Subsequently, in order to reduce the unwanted systematic bias, so that only biologically relevant variations are present in the data and that all samples become comparable in terms of absolute intensities, a normalization step is usually performed [103]. The selection of an appropriate normalization method depends on the type of sample to be analyzed and, over time, several methods have been reported; namely, the use of one or multiple internal standards, total metabolite signal, total protein content, DNA concentration, osmolality, urine creatinine, and probabilistic quotient normalization (PQN), among other algorithms [103,104,105,106]. The main strengths and limitations of each normalization method have been described elsewhere [103,107].

The final preprocessing method is scaling, which considers differences in concentration levels so that changes in abundant metabolites do not dominate statistical models. Thus, scaling methods attempt to normalize the contribution of all variables to the model [108]. Among scaling procedures, the most frequently applied are unit variance (UV), pareto, and logarithmic (log) transformation [96,108].

#### 4.5.2. Multivariate Analysis (MVA)

After the preprocessing steps, multivariate statistical methods are applied to the matrix to reduce the dimensionality of the data due to its ability to deal with several variables simultaneously in a single analysis [96]. MVA is considered the most efficient way to analyze the data, allowing an easier visualization of possible discriminative patterns and thus characterizing the samples based on their metabolic signatures [84]. Multivariate statistical techniques can be divided into two main categories: unsupervised and supervised. Unsupervised techniques, such as principal component analysis (PCA) and hierarchical cluster analysis (HCA), are used to establish intrinsic clusters according to the sample properties without prior knowledge of sample class. Due to its exploratory character, PCA is usually the first approach applied, allowing a rapid identification of similarities and differences between samples and the identification of outliers [109,110]. On the other hand, supervised techniques (such as partial least squares discriminant analysis (PLS-DA) and orthogonal PLS-DA (OPLS-DA)) use prior knowledge about sample class to maximize the separation between two or more sample classes, focusing the analysis on extracting the variables important for group separations [110].

PCA, PLS-DA, and OPLS-DA express their variation into principal components (PCs) or latent variables (LVs), in which each sample is graphically (score plots) expressed as a point (score). Each PC or LV accounts for decreasing proportions of the total variance, and although the number of PCs/LVs can be quite large, generally the first two are sufficient to describe the observed trends [110]. In turn, the contribution of each variable to the variation is shown in the loading plots. A central characteristic between scores and loadings plots is that the direction (positive or negative) on the score plot corresponds to the same direction in the loading plot [111].

When supervised techniques are used for data analysis, the conclusions must be carefully validated due to the risk of data overfitting; while the models (training models) can successfully discriminate groups, they may not be able to classify future samples [112]. Thus, to reduce the overfitting and false discoveries in metabolomic studies, the validation of MVA results is of paramount importance. One of the most common validation methods used in MVA is *n*-fold cross-validation. Cross-validation consists of randomly dividing the dataset into *n* blocks of equal size and then training the model *n* times, while each time keeping out one of the folds as an internal validation set [110,113]. The cross-validation quality parameters R^2^ and Q^2^ are used to provide information about the goodness of the fit (i.e., how well the model explains the dataset) and the predictive capacity (i.e., how well the model is expected to fit additional cohorts), respectively. Thus, an R^2^ around 1 indicates a perfect description of the data by the model, while a Q^2^ around 1 indicates perfect predictability. Generally, a Q^2^ > 0.4 is considered good, and in an ideal model, R^2^ and Q^2^ should be similar [110,113].

Another way to validate the MVA results is by using an independent set of samples (validation set). If the same predictability appears in an independent study, the MVA models can be considered as reliable. In fact, this is the best way to assess the robustness and predictive ability of any model, but this approach is not always possible due to the low number of samples [114]. As alternative, to determine the statistical significance of a model, permutation tests are another great tool [84]. The permutation tests evaluate whether the specific classification of the individuals in the two designed groups is significantly better than any other random classification in two arbitrary groups. In this method, the two class labels are subjected to multiple random rearrangements of the labels on the observed data points. The permutation test is repeated many times, and the permuted and true models are then compared to this distribution of all possible models [115].

#### 4.5.3. Univariate Statistical Analyses

Data analysis methods in metabolomics are mostly based on multivariate analysis, though univariate methods are also used to extract the statistical meaning of the variations observed according to a critical threshold [116]. Although this statistical analysis is easy to use and interpret, it does not take into account the presence of interactions between the different metabolic features, nor the effect of potential confounding variables (e.g., diet), which can increase the likelihood of obtain false positive and/or false negative results [16]. Thus, the combined use of multivariate and univariate data analysis is strongly recommended to maximize the extraction of relevant information from metabolomic datasets [116].

Univariate statistical methods analyze metabolic features independently, based on hypothesis testing, in which a *null hypothesis* (H_0_) postulates a null difference between the mean (or median) of the areas or concentrations of the metabolites detected in the populations under study (e.g., controls and dosed animals). The probability of null hypothesis rejection is calculated (*p*-value), and if it is below the threshold of probability (∝, usually set at 5%), the null hypothesis is rejected [116]. Several univariate statistical tests to compare populations are available. Depending on the statistical data distribution, there are two main families of tests: parametric tests, which are based on the assumption that data are sampled from a Gaussian or normal distribution (e.g., Student’s t-test and ANOVA); and nonparametric tests, which do not make assumptions about the population distribution (e.g., Kruskal–Wallis and Mann–Whitney tests) [116].

#### 4.5.4. The Multiple Testing Problem

In untargeted metabolomic studies, the number of parallel univariate tests depends on the number of *m/z*-RT features detected. Because the number of hypotheses tested is high, as is the probability of incorrectly rejecting a null hypothesis due to random chance, a significant number of false positives (type I errors) may occur, which is particularly undesirable in a metabolomics study [116]. To avoid this problem, which is frequently overlooked by researchers and can jeopardize the results, *p*-values must be corrected. Two possible ways used in metabolomics to deal with multiple testing problems are the Bonferroni and false discovery rate (FDR) corrections. Bonferroni’s correction is extremely conservative, and its use increases type II errors (false negatives), which can result in the loss of many features of interest [116]. In return, FDR tries to maintain a balance, which may be more useful in untargeted metabolomic studies [117].

### 4.6. Metabolites Identification

Translating variables into metabolite identities is one of the main bottlenecks in the untargeted metabolomics approach. Nevertheless, this is an essential prerequisite to integrate data from multiple studies (metabolomic studies or in conjunction with other omics data) and perform an adequate biological interpretation [118]. In most studies, identification is performed using reference spectrum libraries, through appropriate matching criteria. The quality and number of spectrum of metabolites available in these databases are critical to the performance of identification; currently, HMDB [5] is the largest public metabolomics database available. However, this kind of correspondence is only a probable identity assignment, and must be unequivocally confirmed by comparing the retention time/chemical shifts with a pure compound. Mass analyzers with tandem configurations can also assist in the identification of metabolites, since the MS/MS spectra are highly resolved and accurate [116]. In the case of metabolites for which standards are not commercially available, this identification strategy is quite limiting.

Taking into account that the identification process is very time-consuming, it is often carried out only after data analysis for the compounds that have undergone significant changes [118].

### 4.7. Biological Interpretation

After the discriminating metabolites are identified, the potentially affected metabolic pathways should be studied. Several databases are available for this purpose: Kyoto Encyclopedia of Genes and Genomes (KEGG) [119], MetaboAnalyst [120], MetaCyc [121], the small molecule pathways database (SMPDB) [122], and MetaboLights [123], among others. When the metabolite–metabolic pathway association is achieved, a rationale can be elaborated in an attempt to answer the biological question initially formulated.

## 5. Metabolomics: A New Route in Toxicological Research

In the past few years, metabolomics has become a highly versatile tool, creating a new era of opportunities for toxicological research. It has been considered an important area of research that can help to unveil several questions left unanswered by other omics and/or traditional approaches. This topic will focus on the most recent applications of metabolomics in toxicological research.

With increasing demands to reduce drug development time and costs, one of the most important research and development goals in the pharmaceutical industry is the selection of robust new drug candidates with few adverse effects. If a new drug can be screened for adverse toxicity before reaching clinical trials, companies can reduce costs that these trials entail [3]. Through metabolomics, drugs that are likely to fail in late clinical development due to toxicity can be more easily identified in the preclinical development stages, so that the time for the development of new drugs is shortened [124]. Currently, there are several conventional clinical biomarkers of toxicity, including total bilirubin, alanine aminotransferase (ALT), aspartate aminotransferase (AST), alkaline phosphatase (ALP), and creatinine, among others. These biomarkers have limited specificity and sensitivity, since they often only show significantly altered levels after the organs have suffered extensive damage. This sometimes renders preclinical animal studies unable to identify potential toxicity triggered by a new chemical entity, and consequently, they may fail to predict occurrence of toxic effects on the population and subsequent withdrawal of drugs from the market [125]. Therefore, there is currently a great interest in finding novel approaches for the identification of new toxicity biomarkers that are specific indicators of damage in a particular organ. Currently, it has been recognized that metabolomics has enormous potential to detect early toxicity events (e. g. at earlier time points and at lower doses when compared to conventional biomarkers) and identify new toxicity biomarkers, with the majority of studies focusing mainly on renal and hepatic toxicity [125,126]. At the same time, these studies can provide new insights into drug toxicity mechanisms, which is one of the most important aspects of toxicological research [127]. In this regard, Garcia-Canaveras et al. [37] developed an MS-based metabolomics approach to classify and study the different mechanisms of drug-induced hepatotoxicity. For this, they assessed the metabolic profile of human-derived hepatic cells (HepG2) exposed to different hepatotoxic drugs that act by distinct mechanisms, namely steatosis (doxycycline, tetracycline, and valproate), phospholipidosis (amiodarone, clozapine, fluoxetine, tilorone, and tamoxifen), and oxidative stress (cumene hydroperoxide and *tert*-butyl hydroperoxide). Several metabolites have been identified and linked to each toxicity mechanism. Metabolites associated with glutathione and γ-glutamyl cycle were suggested as biomarkers of oxidative stress, and were found altered after exposure to all compounds. In turn, phospholipidosis was characterized by a possible inhibition of phospholipid degradation, while steatosis was related to the increase in triacylglyceride synthesis. Unique metabolomics fingerprints associated with the different mechanisms were used to develop a predictive model with a satisfactory predictive power (area under curve (AUC) of 0.97) for tracking and classifying hepatotoxicity based on the modes of action of the compounds.

In addition to deciphering discriminant mechanism-specific metabolic signatures, it is also possible to identify biomarkers for drug-induced organ toxicity, with several matrices being useful in these kind of metabolomics studies. For example, Shi et al. [128] used NMR to investigate the hepatotoxicity induced by Bay41-4109, an anti-hepatitis B virus compound in rats. After exposure to 10, 50, or 400 mg Bay41-4109/kg for 5 days, urine, serum, and liver tissue were analyzed. In another metabolomics study, Huo et al. [129] used UPLC-MS and NMR as analytical platforms to analyze serum samples of epileptic patients after sodium valproate treatment (0.5 g twice daily for two months) in order to identify diagnostic biomarkers of liver toxicity induced by the mentioned drug. They found differences in the serum metabolic profile of patients with normal liver function and those with elevated liver enzymes, with several metabolites being identified as potential biomarkers, namely glucose, lactate, acetoacetate, acetate, creatinine, very-low-density lipoproteins/low-density lipoproteins (VLDL/LDL), lysophosphatidylcholines (LPCs), phosphatidylcholines, choline, glutamate, alanine, leucine, phenylalanine, *N*-acetylglycoprotein, pyruvate, and uric acid.

Several studies have also been performed in attempts to identify biomarkers of drug-induced nephrotoxicity. Hanna et al. [130] studied changes in the urinary metabolome of newborn rats by GC-MS and LC-MS after administration of gentamicin, an aminoglycoside antibiotic capable of causing acute kidney damage. Three-day-old rats were given a single daily injection of vehicle or gentamicin at doses of 10 or 20 mg/kg/d for 7 days, and urine was collected after the day 3 and day 7 injections. Tryptophan, quinurenic acid, xanthenic acid, and hippuric acid were identified as potential biomarkers for the early detection of acute kidney damage since they were significantly altered 3 days after gentamicin dosage, whereas conventional toxicological biomarkers (serum creatinine and blood urea nitrogen) only revealed significant changes after 7 days. Boudonck et al. [131] provided another example, in which they used an untargeted metabolomics experiment based on GC-MS and LC-MS to study the metabolic changes caused by three proximal tubule nephrotoxic drugs (gentamicin, cisplatin, or tobramycin), and thus established a nephrotoxicity prediction model. For this, urine samples from Sprague Dawley Crl:CD (SD) rats were collected after 1, 5, and 28 days of administration. Increases in polyamines and amino acids were observed in the urine after a single dose of the three compounds, even before histological kidney injury and conventional clinical signs of nephrotoxicity were observed. Upon prolonged administration, nephrotoxic compounds induced a progressive loss of urinary amino acids (leucine, isoleucine, and valine), which allowed the development of a predictive model of nephrotoxicity with high accuracy (70%, 93%, and 100% accuracy at day 1, 5, and 28, respectively) to distinguish nephrotoxic-treated samples from vehicle-control samples.

To a lesser extent, metabolomics has been used to study drug-induced cardiotoxicity and neurotoxicity. One of the first metabolomics experiment in the field of cardiotoxicity was carried out by Andreadou et al. [132] to study the acute toxicity induced by doxorubicin in rats. Three days after doxorubicin administration, aqueous myocardial extracts were analyzed by ^1^H-NMR, revealing that acetate and succinate could be useful as cardiotoxicity biomarkers. Additionally, the authors also showed that oleuropein, a phenolic antioxidant present in the olive trees with documented cardioprotective effects, restored the changes of metabolites to the normal levels. More recently, Li et al. [133] investigated new biomarkers for the evaluation and prediction of cardiotoxicity. Plasma samples of rat cardiotoxicity models in which toxicity was caused by doxorubicin (20 mg/kg), isoproterenol (5 mg/kg), and 5-fluorouracil (125 mg/kg) were analyzed by UPLC-Q-TOF-MS. Metabolomics data revealed 39 biomarkers capable of predicting cardiotoxicity earlier than biochemical and histopathological analysis. However, since drugs with different target organs may cause similar metabolic changes, the identified metabolites were examined in hepatotoxicity and nephrotoxicity models, allowing the researchers to obtain a panel of 10 highly specific biomarkers of cardiotoxicity with a prediction rate of 90%. Among them, L-carnitine, 19-hydroxydioxycortic acid, LPC 14:0, and LPC 20:2 exhibited the strongest specificities for the early prediction of cardiotoxicity.

Studies on drug-induced neurotoxicity are still very limited, although in vitro/in vivo experiments have already demonstrated the benefits and utility of metabolomics to comprehensively detect and characterize neurotoxicity and discover new biomarkers [134,135,136]. For example, in a study conducted by van Vliet et al. [134], rat primary reaggregating brain cell cultures were treated for 48 h with the neurotoxicant methyl mercury chloride (0.1–100 µM) or with the brain stimulant caffeine (1–100 µM), and subsequently, the cellular metabolic profiles were analyzed by LC-MS. The compounds with different modes of action were distinguished by an unsupervised method (PCA), since a treatment-dependent cluster formation was observed, and this effect was concentration-dependent for methyl mercury chloride. Gamma-aminobutyric acid, choline, glutamine, creatine, and spermine have been identified as putative biomarkers for methyl mercury chloride neurotoxicity. In addition, the authors also assessed the metabolic alterations induced by subcytotoxic concentrations (1 µM) of eight compounds (trimethyltin chloride, methyl mercury chloride, colchicine, paraquat, cycloheximide, dimethylformamide, dichlorophenoxy acetic acid, and acetaminophen) with specific target organ toxicity for the brain, liver, and kidneys through different mechanisms of toxicity. The PCA revealed cluster formations that were largely dependent on target organ toxicity, indicating the possibility of developing a neurotoxicity prediction model.

Toxicometabolomics has grown far beyond its initial preclinical and clinical applications. A rising number of studies showed that metabolomics is a powerful tool for elucidating biochemical modes of action and toxicological effects of a wide range of toxic compounds in a variety of toxicology domains, including food safety [137,138], environmental toxicology [139,140], and forensic toxicology [141,142]. Metabolomics has also recently gained attention as a novel tool in regulatory toxicology [143]. Table 3 summarizes some other recent examples of toxicometabolomic studies, grouped by the following specific research objectives: (a) elucidation of toxicity mechanisms; (b) construction of toxicity prediction models; and (c) identification of toxicity biomarkers.

## 6. Current Challenges and Future Perspectives

Toxicological research has made significant advances since the advent of metabolomics. Over the last few years, a large amount of data has been generated using metabolomic approaches, resulting in the identification of several potential biomarkers associated with susceptibility, exposure, and/or effects of toxicants, and a better understanding of the metabolic responses of many biological systems has been achieved.

Given the lack of standardized metabolomic protocols, some toxicometabolomic studies using similar toxic agents, species, and biological specimens resulted in disparate conclusions. To further increase the reproducibility and translatability of metabolomic studies, it is urgent to standardize the different steps of the metabolomic workflow, including the experimental design and statistical analysis, as these steps can considerably affect the obtained results. It should also be noted that the use of different models, which are frequently simplistic, as is the case of in vitro models, as well as the use of small cohorts, are potential sources of bias. Furthermore, despite the high potential of metabolomic approaches in hypothesis generation and biomarker discovery, the translation of these findings to the real world remains low. Preliminary metabolomic results must be validated, and for that, putative biomarkers identified through metabolomics studies must be accurately and precisely measured in larger groups of individuals. Special consideration should also be given to data stored in databases and biobanks, which must be significantly expanded and improved.

The use of metabolite ratios to successfully compare datasets from different studies is another important recommendation to take into account in future studies, as the ratios between related metabolite pairs reduce overall noise and biological variability in the dataset. More importantly, metabolite ratios show the flux through a metabolic pathway when a metabolite pair is coupled by that pathway. However, measuring metabolic flux with stable isotope tracers is equally important, and has already been used successfully in metabolomics research. Tracers based on stable isotopes can reveal which parts of a metabolic network are in use.

In conclusion, while there are still some bottlenecks in metabolomics, it is expected that with continuous optimization and improvement of research methods, the use of metabolomics to uncover previously unknown information will become more accurate and efficient, creating a powerful tool for improving human health and environmental safety.

## Figures and Tables

**Figure 1 metabolites-11-00692-f001:**
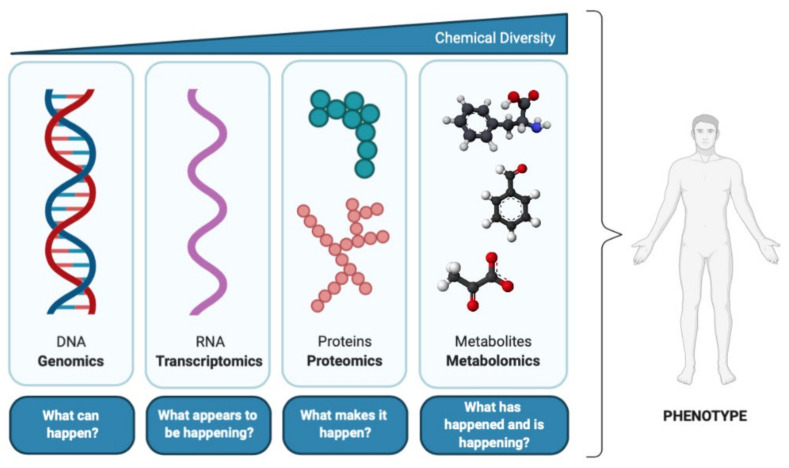
The central dogma of molecular biology and correspondence with ‘omics’ disciplines (adapted from [17]).

**Figure 2 metabolites-11-00692-f002:**
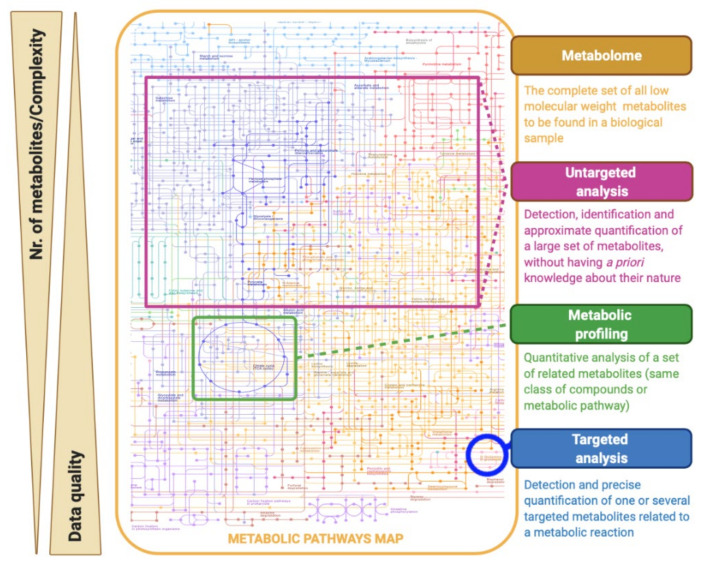
Classification of different strategies of metabolomic experiments (adapted from [22]).

**Figure 3 metabolites-11-00692-f003:**
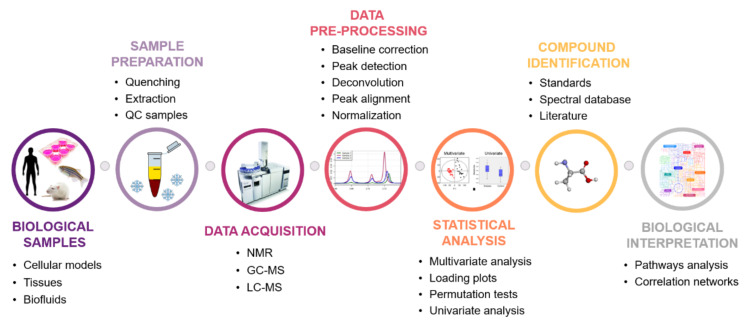
Schematic representation of a general untargeted metabolomics workflow.

**Table 1 metabolites-11-00692-t001:** Advantages and limitations of different omics technologies (adapted from [21]).

Omics Technology	Strengths	Limitations
**Genomics**	- High-throughput sequencing techniques allow the cost and time-efficient sequencing of complete genomes; - Simultaneous gene expression analysis of thousands of genes; - Sensitive endpoint of toxicity, since gene expression changes often occur at an early stage; - Studies on gene polymorphisms clearly demonstrate the individual susceptibility to some drugs and different responses among individuals.	- Due to polymorphism, genome sequencing alone is not enough; - Changes in gene expression do not always lead to adverse effects; - Difficult to predict the final biological effect of DNA by only genome analysis due to post-transcriptional, post-translational, and epigenetic changes; - Difficult to translate results to in vivo toxicity or diseases.
**Transcriptomics**	- Vast quantity of data is produced; - Effective combination with single cell technologies; - Relatively inexpensive; - Good reproducibility for interlaboratory studies.	- RNA isolation and sequencing are susceptible to handling errors; - Presence of RNAs does not necessarily predict the translation into proteins; - Does not take into account post-transcription modifications; - Need for specialized analysts.
**Proteomics**	- Sensitivity, specificity, and low costs of protein arrays; - Simultaneous analysis of thousands of proteins; - Analysis of protein–protein interactions; - Allow quantitative analysis; - Robust link between an organism proteomic profile and its phenotype.	- Complexity and instability of the proteome; - Large number of proteins and possible post-translational modifications; - Limited reproducibility; - Laborious analysis technique; - Not all proteins in a sample can be identified; - Expensive equipment is required.
**Metabolomics**	- Simultaneous analysis of a large set of metabolites; - Considered to be closest to the phenotype; - Availability of a public and commercial database; - Chemical structures of the metabolites are universal, allowing translatability between species; - Diverse range of applications across many fields.	- Metabolome is influenced by several variability factors; - Sample collection and preparation conditions, as well as analytical platforms chosen, limit the detection of some metabolites; - Require expensive analytical techniques; - Complexity of the data analysis and interpretation.

**Table 2 metabolites-11-00692-t002:** A summary of the most relevant characteristics of the analytical techniques commonly used in metabolomics [1,16,63,85,86,87,91,92,93].

Analytical Technique		Strengths	Weaknesses
**Nuclear magnetic resonance (NMR)**		- Rapid analysis (a few minutes); - All types of small molecules can be measured simultaneously; - High accuracy and repeatability; - Liquid and solid matrices; - Nondestructive; - Minimal sample preparation; - Detailed structural information; - Allows the quantification of all metabolites using a single internal standard; - Enables high-throughput measurements.	- Low sensitivity (limit of detection ~1 µM) - Limited resolution; - Detects fewer metabolites compared to MS; - More than one peak per metabolite in most cases, which means that spectra are often complex; - Libraries of limited use due to complex matrix; - NMR spectrometers are very expensive and take up a lot of space; - Requires skilled technicians.
**Mass spectrometry (MS)**	**Direct infusion-mass** **spectrometry (DI-MS)**	- Fast and highly reproducible; - Requires a small amount of sample; - No loss of metabolites in sample preparation; - High-throughput detection; - Data processing is relatively simple.	- Does not allow quantification; - Ion suppression/ion enhancement; - Low ionization efficiencies.
	**Capillary electrophoresis-mass spectrometry (CE-MS)**	- Small volume of samples; - Short analysis time; - Minimal sample preparation; - Wide variety of molecules can be analyzed, including thermolabile compounds.	- Poor reproducibility; - Low sensitivity; - Affected by salts in the sample; - Less stable than LC-MS and GC-MS.
	**Liquid chromatography-mass spectrometry (LC-MS)**	- Very high sensitivity (< µM); - Robust; - Enables analysis of thermolabile metabolites; - Simple sample preparation; - Suitable for the study of lipids, di- and tripeptides, and other macromolecules.	- Ion suppression/ion enhancement; - Destructive; - Sample analysis can be long (20–60 min/sample); - More instrumental variables; - High solvent consumption; - Novel compounds identification can be difficult.
	**Gas chromatography-mass spectrometry (GC-MS)**	- Very high sensitivity (< µM); - Large linear range; - Enables simultaneous analysis of different classes of metabolites; - Compound identification is facilitated by mass spectral libraries.	- Extensive sample preparation; - Destructive; - Limited to volatile compounds or that require prederivatization processes; - Sample analysis can be slow; - Novel compounds identification can be difficult.
	**High-resolution mass** **spectrometry (HRMS)**	- Enables the determination of accurate mass and isotopic distribution; - High resolution, selectivity, and specificity; - Useful in metabolite identification; - Method development for a quantification assay can be faster.	- High costs; - Complex instrument maintenance; - Data analysis can be complex.

**Table 3 metabolites-11-00692-t003:** Representative examples of recent studies using metabolomics in toxicological research.

Toxicant Classification/ Name	Study Model/ Matrix	Study Design	Analytical Platform	Study Outcomes	Ref.
**A. ELUCIDATION OF TOXICITY PATHWAYS**					
Antidepressant drug: Venlafaxine (VEN)	In vitro model: Primary rat astrocytes	Astrocytes were treated with 10 µM VEN (*n* = 30), or with DMSO as a vehicle control (*n* = 30) for 72 h. The intracellular metabolites were profiled.	^1^H NMR	Metabolic pathways significantly disturbed: Amino acid metabolism Glycolysis Cholesterol metabolism	[144]
Antineoplastic drug: Doxorubicin (DOX)	Animal model: Plasma of male Wistar rats	Rats were randomly divided into a treatment group injected i.p. with 3 mg DOX/kg once a week, for 6 weeks (*n* = 9), and a control group injected i.p. with saline (*n* = 8)	UPLC-Q-TOF-MS	Metabolic pathways significantly disturbed: Phenylalanine, tyrosine, and tryptophan biosynthesis D-glutamine and D-glutamate metabolism Phenylalanine metabolism Biosynthesis of unsaturated fatty acids	[145]
Opioid analgesic: Tramadol	Animal model: Cerebrum of Kunming mice	Mice were treated with 0, 20, or 50 mg tramadol/kg/day, via oral gavage for 5 weeks (*n* = 6/group)	GC-TOF-MS	Metabolic pathways significantly disturbed: By low dose of tramadol: - Valine, leucine, and isoleucine degradation - Galactose metabolism By high dose of tramadol: - Sphingolipid, fructose, and mannose metabolism	[146]
Drug of abuse: MDMA	In vitro model: Primary mouse hepatocytes	Hepatocytes were exposed to subtoxic (LC_01_: 203 µM and LC_10_: 472 µM) and toxic concentrations (LC_30_: 757 µM) for 24 h (*n* = 10/group). The intracellular metabolites were profiled.	GC-MS	Metabolic pathways significantly disturbed: Amino acid metabolism Aminoacyl tRNA biosynthesis Glutathione metabolism TCA cycle Pyruvate metabolism	[147]
Drug of abuse: MDPV	Animal model: Organs (liver, kidney, heart, and brain) and urine of male CD-1 mice	Mice were exposed to human-relevant doses (3 × 2.5 mg/kg and 3 × 5 mg/kg, i.p.) and sacrificed 24 h after the first administration (*n* = 10/group)	GC-MS	Metabolic pathways significantly disturbed in: Liver: - Cysteine and methionine metabolism - Biosynthesis of valine, leucine, isoleucine, and unsaturated fatty acids - Nitrogen and glutathione metabolism - Aminoacyl-tRNA and pantothenate biosynthesis Kidney: - Nitrogen and glutathione metabolism - Aminoacyl-tRNA and pantothenate biosynthesis - Cyanoamino acid metabolism - Phenylalanine, tyrosine, and tryptophan biosynthesis - Methane, glycine, serine, threonine, beta-alanine, alanine, aspartate, and glutamate metabolism Heart: - Fatty acid biosynthesis Brain: - Butanoate metabolism - Synthesis and degradation of ketone bodies - Amino acids and nitrogen metabolism - Aminoacyl t-RNA biosynthesis Urine: - Starch and glucose metabolism - Glutathione metabolism	[126]
Heavy metal (environmental pollutant): Cadmium (Cd)	Human model: Urine	144 volunteers (*n* = 99 females and *n* = 45 males) living in three nearby villages with different levels of Cd: control area (<0.05 mg/kg), low-polluted area (0.2–0.4 mg/kg), and high-polluted area (>0.4 mg/kg) (according to the Cd content found in rice and vegetables growing in the area)	UHPLC-Q-Exactive Orbitrap MS	Metabolic pathways significantly disturbed: Creatinine pathway Tryptophan metabolism Aminoacyl-tRNA biosynthesis Purine metabolism	[148]
Heavy metal (environmental pollutant): Cd	Animal model: Urine of wild-type 129/Sv female mice	Mice were fed with 300 ppm Cd-containing chow (*n* = 5) for 67 weeks and compared to the control group (*n* = 4)	UPLC-QTOF-MS	Metabolic pathways significantly disturbed: Arginine and proline metabolism Alanine, aspartate, and glutamate metabolism Aminoacyl-tRNA biosynthesis Purine metabolism	[149]
**B. IDENTIFICATION OF TOXICITY BIOMARKERS**					
Hepatotoxic agent: Hydrazine	Animal model: Serum and urine of male Wistar rats	Rats were randomly divided into four groups: two control groups (*n* = 12/group) and two hydrazine-treated groups (*n* = 18/group). One control group and one hydrazine-treated group were allocated for sampling at 24 h postdosing, while the remaining two groups were for sampling at 48 h postdosing. The hydrazine-treated groups were orally administrated with a single dose of hydrazine (150 mg/kg), at which hydrazine could induce an obvious histopathological effect and hepatocellular lipid accumulation.	RRLC-MS	A biomarker group was proposed, including 6 upregulated (creatine, tryptophan, *N*-acetylhistidine, l-carnitine, pyroglutamic acid, and indole acrylic acid) and 10 downregulated (proline betaine, l-acetylcarnitine, pipecolic acid, xanthurenic acid, trigonelline, kynurenic acid, indole-3-carboxylic acid, phosphorylcholine, 4-pyridoxic acid, and thymine) metabolites with AUC > 0.85 The biomarker panel provided an AUC of 1 and a specificity and sensitivity of 100%	[150]
Neurotoxic agent: Sevoflurane (SEVO)	Animal model: Serum samples of offspring rats born from maternal Sprague Dawley rats	Rats within 18–19 days of gestation were randomly divided into control (CTR) or sevoflurane (SEVO) groups (*n* = 4/group). In the SEVO group, animals were treated with 2% sevoflurane carried by 100% oxygen for 6 h. For the control group, animals were placed in an identical condition without sevoflurane. Then, 26 postnatal-7-day rats were randomly selected from offspring generation groups (*n* = 13/group) and decapitated, and samples were collected for metabolomic analysis.	UPLC-TOF-MS	S-Adenosylmethioninamine, DG(14:0/0:0/22:4n6), DG(20:3(8Z,11Z,14Z)/16:0/0:0), allantoin, DG(16:0/0:0/20:3n9), methylsuccinic acid, cholic acid, cervonoyl ethanolamide, DG(20:0/0:0/18:3n6), (R)−1,2-dimethyl-5,6 dihydroxytetrahydroisoquinoline, 11b-PGF2a, 5-aminopentanoic acid, porphobilinogen, proline, methionyl-proline, oleanolic acid, docosapentaenoic acid, 3-hydroxykynurenine, 2-methoxybenzoic acid, leucyl-lysine, hydroxyprolyl-proline, leukotriene E4, calcitroic acid, 8,11,14-eicosatrienoic acid, 5b-cyprinol sulfate, lysoPC(22:5(4Z,7Z,10Z,13Z,16Z)), sulfolithocholylglycine, TG(22:0/22:6), and 12-ketodeoxycholic acid were identified as potential neurotoxicity-related biomarkers Correlation of serum metabolomic data with hippocampus enzyme-linked immunosorbent assay suggested that S-adenosylmethioninamine was the most important biomarker of prenatal exposure to sevoflurane	[151]
Cardiotoxic agent: Cyclophosphamide (CY)	Animal model: Plasma of male Wistar rats	Rats were randomly divided into four groups: control (*n* = 15), CY-1d, CY-3d, and CY-5d groups (*n* = 10/group). CY was administered i.p. to the rats on the first day, and the dosage was set at 200 mg/kg. The control group was administered i.p. with 1 mL saline on the first day. Rat plasma samples were collected one, three, and five days after CY administration.	UPLC-QTOF-MS	16 metabolites were found significantly altered in CY groups (L-carnitine, proline, 19-hydroxydeoxycorticosterone, phytosphingosine, cholic acid, LPC (14:0), LPC (18:3), LPC (16:1), LPE (18:2), LPC (22:5), LPC (22:6), linoleic acid, LPC (22:4), LPC (20:2), LPE (18:0), and LPC (20:3) A relationship between plasma metabolomic data and heart biochemistry and histopathologic changes suggested that the selected metabolites could act as sensitive biomarkers for CY-induced cardiotoxicity	[152]
Nephrotoxic agent: Cisplatin	Animal model: Plasma and kidney tissue of male Crl:CD (SD) rats	Rats were randomly divided into three groups: the high-dose group (10 mg cisplatin/kg i.p., *n* = 13), low-dose group (5 mg/kg of cisplatin i.p., *n* = 10), and untreated group (*n* = 10). Blood samples were collected from the tail vein at 24, 48, and 96 h after the injection of cisplatin. Rats were sacrificed at 96 h, and kidney samples were also harvested for the metabolome analysis	GC-MS and LC-MS	Cysteine-cystine and 3-hydroxy-butyrate (based on GC/MS analysis) and AC 14:0, AC 18:1, AC 18:2, and PE 18:2–18:2 (based on LC/MS analysis) were found both in plasma and kidney tissue, and were identified as candidate biomarkers to detect cisplatin-induced nephrotoxicity early	[153]
Nephrotoxic agent: Gentamicin (GM)	Animal model: Serum and urine of male Sprague Dawley rats	Rats were given 0, 30, or 300 mg GM/kg/day i.p. for 3 consecutive days (*n* = 4–5/group) and were sacrificed 2 days (D2) or 8 days (D8) after last administration.	^1^H NMR	Five serum metabolites (3-hydroxybutyrate, citrate, creatine, glucose, and glycine) were selected, based on D2 and D8 results, as biomarker for nephrotoxicity Nine urinary metabolites (2-oxoglutarate, acetate, citrate, glucose, glycine, hippurate, lactate, succinate, and taurine) were also selected, based on D2 and D8 results, as biomarkers for nephrotoxicity Correlation of serum and urinary ^1^H NMR OPLS-DA with serum biochemistry and renal histopathologic changes suggested that the selected biomarkers may be used to reliably predict or screen for GM-induced nephrotoxicity	[154]
Herbicides: Metribuzin, glyphosate and their mixtures	Aquatic plant model: *Lemna minor* L.	Plants were exposed for 72 h to concentrations of metribuzin or glyphosate equal to their corresponding EC_50_ values, or their mixtures (25–75, 50–50, or 75–25% of their corresponding EC_50_ values) (*n* = 6/group).	GC/EI/MS	Identification of GABA, salicylate, caffeate, α,α-trehalose, and squalene as a set of biomarkers useful in the evaluation of *Lemna* stress levels caused by herbicides Salicylate was identified as a specific biomarker of the toxicity caused by metribuzin/glyphosate mixtures	[155]
Heavy metal: Arsenite	Animal model: Zebrafish (ZF) embryos	ZF embryos were exposed to sodium arsenite under different concentrations (0.5, 1.0, 2.0, and 5.0 mg/L) 24, 48, and 72 h postfertilization. ZF embryonic homogenate was used for metabolomic analysis.	UPLC-QTOF-MS	A group of 10 metabolites was able to discriminate the arsenite and control groups with 99% accuracy, being identified as potential biomarkers for arsenic exposure in early development life stages These 10 metabolites were 9-hydroxy-10-O-D glucuronoside-12Z-octadecenoate, vinaginsenoside R3, PG(18:0/22:6(4Z,7Z,10Z,13Z,16Z,19Z)), methyl (3x,4E,10R)-3,10-dihydroxy-4,11-dodecadiene-6,8-diynoate 10-glucoside, hexaethylene glycol, N1-(2-methoxy-4-methylbenzyl)-n2-(2-(5-methylpyridin-2-yl)ethyl) oxalamide, butyl (*S*)-3-hydroxybutyrate glucoside, ganglioside GM2 (d18:0/22:1(13Z)), indanone, and estrange.	[156]
**C. CONSTRUCTION OF TOXICITY PREDICTION MODELS**					
Drugs from several different therapeutic classes with cardiotoxic potential	In vitro model: Pluripotent stem cell-derived cardiomyocytes (hiPSC-CM)	Phase 1: 66 drugs tested at a single, noncytotoxic concentration were used to identify predictive metabolites that could discriminate cardiotoxicants from noncardiotoxicants independent of changes. Phase 2: the discriminatory metabolites identified in phase 1 were used to create an exposure-based, targeted assay for identifying a drug’s cardiotoxicity potential. The predictivity was evaluated with 81 drugs (52 cardiotoxic and 29 noncardiotoxic).	UPLC-HRMS	Four metabolites that represent different metabolic pathways (arachidonic acid, lactic acid, 2′-deoxycytidine, and thymidine) were identified as indicators of cardiotoxicity A cardiotoxicity predictive model with 85% balanced accuracy, 90% sensitivity, 79% specificity, 89% PPV, 82% NPV, and an AUC of 0.887 was developed based on the four discriminatory metabolites	[157]
Cardiotoxic drugs: DOX, isoproterenol (ISO) and 5-fluorouracil (5-FU)	Animal model: Plasma of male Wistar rats	Phase 1: 100 rats were randomly divided into 10 groups (*n* = 10/group) to screen the potential biomarkers for the early prediction of cardiotoxicity. For each drug, different doses and sampling times were tested. Phase 2: 70 rats were randomly divided into seven groups, which included control, two cardiotoxicity groups (ISO and 5-FU), two hepatotoxicity groups (Radix Bupleuri and carbon tetrachloride), and two nephrotoxicity groups (gentamicin and etimicin), to examine the specificity of the selected biomarkers. Phase 3: the discriminatory metabolites selected in phase 2 were used to create a predictive model of drug-induced cardiotoxicity in its early stages.	UPLC-Q-TOF-MS	The predictive model that combined L-carnitine, 19-hydroxydeoxycorticosterone, LPC (14:0), and LPC (20:2) exhibited the strongest specificities. The prediction rate was 90%	[133]
Two overt hepatotoxicants (acetaminophen (APAP) and carbon tetrachloride (CCl_4_)), two idiosyncratic hepatotoxicants (felbamate (FEL) and dantrolene(DAN)), and three nonhepatotoxicants (meloxicam (MEL), penicillin (PEN) and metformin (MET))	Animal model: Blood of male Sprague Dawley rats	Rats were orally gavaged with a single dose of vehicle (*n* = 4 for APAP study and *n* = 5 for other compound studies), low dose or high dose of the compounds (100 or 1250 mg APAP/kg, 50 or 2000 mg CCl_4_/kg, 300 or 1920 mg FEL/kg, 100 or 1000 mg DAN/kg, 100 or 1500 mg MET/kg, 0.4 or 12 mg MEL/kg, and 100 or 2400 mg PEN/kg (*n* = 7/APAP groups and *n* = 5/for all other groups)). At 6 and 24 h postdosing, blood was collected for metabolomics analysis.	LC-QTOF-MS	From the studies with APAP and CCl_4_, 41 metabolites were selected to build models to predict hepatotoxicity. PLS modeling results showed 89% accuracy in the modeling set This model was further used to predict the response of rats treated with non- or idiosyncratic hepatotoxicants at 6 and 24 h A model with an accuracy of at least 97.4% for the hold-out test set and 100% for training sets was developed	[158]

Abbreviations: AC, acylcarnitine; AUC, area under curve; DG, diglyceride; GABA, gamma aminobutyric acid; GC-TOF-MS, gas chromatography time-of-flight mass spectrometry; GM2, disialotetrahexosylganglioside; i.p., intraperitoneal; LPC, lysophosphatidylcholine; LPE, lysophosphatidylethanolamine; NPV, negative predictive value; PE, phosphotidylethanolamine; PG, phosphatidylglycerol; PPV, positive predictive value; RRLC, rapid resolution liquid chromatography; UPLC-HRMS, ultra-high-performance liquid chromatography–high-resolution mass spectrometry; UPLC-Q-TOF-MS, ultra-high-performance liquid chromatography-quadrupole time-of-flight mass spectrometry.

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
