# Peer review of "Toxicometabolomics: Small Molecules to Answer Big Toxicological Questions"

_metabolites, 2021, doi:10.3390/metabo11100692_

Round 1
Reviewer 1 Report
This review is about toxicometabolomics, an area of toxicology, which still has a lot to exploit. I really liked how the authors describe in detail and clearly and concisely, each term, and meaning of this area of science. From the introduction, the concepts and origin of this area are clarified and compared to other sciences (genomics, proteomics ...) which allows the reader to directly understand the interest of toxicometabolomics. All well based on a detailed bibliography, and with adequate English. The metabolomics workflow is very interesting, since it details each process from the questions to be asked before starting the experimentation process, as well as the selection of the samples, how the samples are treated, the analytics, but also the most complex part , treat the results from the statistical point and all the variables that it entails. The use of probability is also discussed, since when dealing with so many variables it is important to evaluate it, and here the authors explain it with the current possibilities. From the beginning to the end, all the advantages and limitations of each stage are discussed, comparing the techniques currently used. Tables and figures also allow the information to be viewed and structured in an appropriate and clear way.
I can only say that it has been a pleasure to read this review, and I only have a few comments or questions to ask:
-Line 191: the authors talk about new models that increase the parameters to be controlled, which ones, for example?
Line 256: the preservative sodium fluoride, also widely used, is not mentioned. What are its advantages and limitations?
Line 356: it speaks of ion suppression, however there is also the ‘ion enhancement’ by increasing the nominal value. I think it would be more appropriate to mention both possibilities.
Table 2: liquid chromatography / mass spectrometry: is TOF included here? because there are good libraries for TOF. Long analytical time: it is relative, there are short methods of less than 20 min. Ion suppression / ion enhancement (I would include both).
Reviewer 2 Report
This review describes the basics and few applications of metabolomics to toxicological studies. The aim is very ambitious, considering the vastity of the topic. Therefore, eventually the manuscript covers every aspect but it fails to be exhaustive of the principles and the existing literature. However, it can be considered acceptable as a review for those who are about to approach metabolomics studies. The paragraph 4.4 and Table 2 should include HRMS
Reviewer 3 Report
The article "Toxicometabolomics: Small Molecules to Answer Big Toxicological Questions" is a very well done review, with updated data that includes the latest advances in Metabolomics. The introduction is adequate and concise, and the tables are helpful in illustrating and summarizing the state of the art. Relevant authors have been included and a very complete description of the currently most relevant advances in analysis and techniques has been made. The table on representative examples in toxicological research is particularly interesting
Reviewer 4 Report
This is a well-written review that provides an informed overview of metabolomics itself and how the science fits into the field of biomarker identification and hypothesis development. The review provides a context for how toxicometabolomics contributes to drug development and toxic response identification and mechanisms.
A few small suggestions:
- In section 4.2.1 Cellular models, when discussing the use of primary cells, it should be noted that some of the limitations of their use are the loss of viability that many undergo upon isolation and placement in culture and that the conditions of their isolation can affect their overall responses.
- In section 5 Metabolomics: a new route in toxicological research, when discussing it would be useful to state the sensitivity of the metabolomics biomarkers as predictive biomarkers compared to the traditional markers of liver or other organ-specifici toxicity (i.e. earlier time of detection, lower dose, etc.)
- Section 6 Current challenges and future perspectives can be expanded to include discussions of use of metabolite ratios as biomarkers. Also address the differences in experimental design and species for in vivo studies and how this can affect predictive model development.
